# Autoantibodies in Atrial Fibrillation—State of the Art

**DOI:** 10.3390/ijms24031852

**Published:** 2023-01-17

**Authors:** Joanna Zygadło, Grzegorz Procyk, Paweł Balsam, Piotr Lodziński, Marcin Grabowski, Aleksandra Gąsecka

**Affiliations:** 11st Chair and Department of Cardiology, Medical University of Warsaw, Banacha 1A, 02-097 Warsaw, Poland; 2Doctoral School, Medical University of Warsaw, 02-091 Warsaw, Poland

**Keywords:** atrial fibrillation, autoantibodies, arrhythmias, autoimmunization

## Abstract

Atrial fibrillation (AF) is the most common type of cardiac arrhythmia. To date, a lot of research has been conducted to investigate the underlying mechanisms of this disease at both molecular and cellular levels. There is increasing evidence suggesting that autoimmunity is an important factor in the initiation and perpetuation of AF. Autoantibodies are thought to play a pivotal role in the regulation of heart rhythm and the conduction system and, therefore, are associated with AF development. In this review, we have summarized current knowledge concerning the role of autoantibodies in AF development as well as their prognostic and predictive value in this disease. The establishment of the autoantibody profile of separate AF patient groups may appear to be crucial in terms of developing novel treatment approaches for those patients; however, the exact role of various autoantibodies in AF is still a matter of ongoing debate.

## 1. Introduction

Autoimmune diseases, such as type 1 diabetes mellitus, rheumatoid arthritis, and psoriasis, derive from the failure of normal self-tolerance mechanisms. Although autoantibodies have been proven to be the cause of many diseases, factors that are involved in their production and their exact role in pathogenesis remain largely unresolved [1]. 

Autoimmune diseases affect about 5–7% of the population and they have a significant contribution to the general mortality and morbidity [2]. The number of illnesses attributed to autoimmunity is still growing as new findings are being discovered in ongoing trials. 

Recently, autoimmunity has been evidenced to be a contributor, a cause, and a predictive factor of cardiovascular system diseases such as cardiomyopathies, myocarditis, heart failure, and atherosclerosis [3,4,5]. Lately, many research groups have investigated the role of autoimmunity in the development of cardiac arrhythmias including atrial fibrillation (AF) [6,7,8,9]. 

Atrial fibrillation is the most prevalent sustained cardiac arrhythmia in adults [10]. In the United States the lifetime risk of AF occurrence was estimated to be approximately 1 in 3 among White People, and 1 in 5 among Black People [11]. Currently, the prevalence of AF in adults ranges between 2% and 4%, and a 2.3-fold rise by 2030 is expected [12,13]. Atrial fibrillation is related to an increased risk of stroke and peripheral embolism. Moreover, it is associated with an increased mortality rate. Many independent risk factors for AF occurrence have been identified so far in population-based cohort studies including e.g., diabetes, hypertension, congestive heart failure, and valve disease [14]. Nevertheless, the pathophysiological mechanisms of AF remain unclear despite detailed research in this field [15,16]. Extensive clinical and scientific evidence demonstrates that inflammatory mechanisms play an important role in promoting AF [17,18]. Many studies have highlighted the role of electrical and structural changes in the atria in the initiation and maintenance of AF. These changes include ion channels remodeling, atrial fibrosis, and altered autonomic tone [19,20].

The search for biomarkers enabling diagnosis, prognosis, and treatment response prediction in different cardiovascular diseases has included not only various microparticles, such as micro-ribonucleic acids or extracellular vesicles, but also autoantibodies [21,22,23]. Autoantibodies have been found to target various cardiac receptors modulating the impact of the autonomic nervous system on signaling pathways [7,24]. Autoantibodies can bind to the extracellular receptor causing a direct change in the protein function, but they can also lead to cell death in a complement-mediated manner. Growing evidence suggests that autoantibodies play a pivotal role in the regulation of heart rhythm and conduction system and, therefore, are associated with AF development [6,25,26] (Figure 1). In this review, we have summarized the current knowledge concerning the role of autoantibodies in AF development as well as their prognostic and predictive value in this disease.

## 2. Autoantibodies in Patients Suffering from Atrial Fibrillation

We searched the PubMed Database and eventually included 31 original research publications (both clinical and preclinical) relevant to the discussed field, excluding only the records that were found by coincidence. We divided these studies into the following groups: (i) research investigating only antibodies against β-adrenergic receptors (anti-β antibodies), (ii) research investigating only antibodies against M2-muscarinic acetylcholine receptor (anti-M2 antibodies), (iii) research investigating both anti-β and anti-M2 antibodies, (iv) research investigating anti-heat shock protein (anti-HSP) antibodies, (v) research investigating other types of autoantibodies.

### 2.1. Research Investigating Only Anti-β Antibodies

Novikova et al. studied patients with arrhythmias of various etiology. The group of patients was further subdivided into: (i) primary (idiopathic) electrical abnormalities patients, (ii) chronic myocarditis (CM) and dilated cardiomyopathy (DCM) patients, and (iii) ischemic heart disease (IHD) patients. They evidenced that there was an increased level of anti-β1 antibodies in ventricular tachycardia, ventricular extrasystole or AF patients (excluding patients with IHD) compared to controls [27].

Another research team evaluated the C-reactive protein (CRP) levels and the incidence of anti-β1 antibodies in patients suffering from supraventricular tachyarrhythmias. They found that patients with arrhythmias (including AF) showed a significantly higher prevalence of anti-β1 antibodies than controls. Moreover, patients with supraventricular arrhythmias (along with DCM or CM) had higher median CRP values than patients with idiopathic arrhythmias, those with supraventricular arrhythmias with coexisting arterial hypertension (AH), and healthy controls [28]. 

Nikitina et al. investigated the prevalence of anti-β1 antibodies and the heart rate variability (HRV) in 42 arrhythmic patients subdivided into two groups: (i) idiopathic arrhythmias patients: 13 paroxysmal AF or flutter, two paroxysmal atrial tachycardia (PAT), and 16 paroxysmal ventricular tachycardia (PVT), (ii) 11 patients suffering from PVT with coexisting DCM or CM. An increased occurrence of anti-β1 antibodies was found in arrhythmic patients as compared to healthy controls. Additionally, within the idiopathic arrhythmic patients’ group, it was demonstrated that patients with anti-β1 antibodies had decreased HRV parameters compared to anti-β1 antibodies negative patients. These findings indicate that anti-β1 antibodies might contribute to the dysfunction of chronotropic heart regulation and might lead to arrhythmias evolution [29].

Interestingly, it was proved that immunizing rabbits with β2-adrenergic receptors (β2) caused atrial arrhythmias mainly in the form of sustained atrial tachycardia [30]. Similarly, Li et al. induced sustained atrial tachycardia in rabbits by immunizing them with β2 with the following injection of thyroxine (T4). On the other hand, rabbits immunized with β1-adrenergic receptors (β1) and injected with T4 presented with sustained sinus tachycardia [31]. Noteworthily, T4 may exert the immune response (both physiological and pathological) including the activation of lymphocytes and thus further increasing autoantibody levels.

Shang et al. studied anti-β1 antibody levels relative to left atrium diameter (LAD) and circulating fibrosis biomarker levels in a group of 70 patients with paroxysmal AF. The anti-β1 antibody levels were found to be positively correlated with LAD and circulating fibrosis biomarker levels. Furthermore, they established a rabbit model overexpressing anti-β1 antibodies. It was discovered that anti-β1 antibodies increased AF inducibility by facilitating atrial fibrosis, transforming growth factor-β1 (TGF-β1) signaling activation, and collagen accumulation [32]. All studies discussed in this section have been summarized in Table 1.

### 2.2. Research Investigating Only Anti-M2 Antibodies

Baba et al. demonstrated that the presence of anti-M2 antibodies was the only independent predictor of AF in DCM patients. Moreover, they conducted an experiment in which chick embryos were injected with IgG derived from: (i) DCM patients anti-M2 positive, (ii) idiopathic AF patients anti-M2 positive, (iii) healthy controls, (iv) DCM patients anti-M2 negative, (v) idiopathic AF patients anti-M2 negative. It was shown that purified anti-M2 IgG from both AF and DCM patients exhibited a negative chronotropic effect - indicating that anti-M2 autoantibodies could affect sinus node function and induce supraventricular arrhythmias [33].

Another group evaluated the effects of immunizing rabbits with a synthetic peptide corresponding to the M2 with further assessment of atrial electrophysiology in isolated perfused rabbit hearts. Immunized rabbits produced anti-M2 antibodies and showed a significantly shorter atrial effective refractory period and a longer intra-atrial activation time compared to control rabbits. Furthermore, they showed overexpression of the M2-receptor-I_K,Ach_ (muscarinic-gated potassium channel) pathway, and increased atrial fibrosis. Taken together, it was demonstrated that anti-M2 antibodies were able to induce atrial remodeling in terms of both structure and electrophysiology, which pointed out that these antibodies might participate in the induction and perpetuation of AF [34].

Zou et al. studied the predictive value of preprocedural anti-M2 antibodies levels for the recurrence of AF in a group of 76 AF patients (paroxysmal or persistent) with preserved left ventricular ejection fraction who were enrolled for the ablation procedure. In patients with AF, the frequency and levels of the anti-M2 antibodies were higher than those in patients with sinus rhythm. Persistent AF patients had higher frequency and titers of these antibodies when compared to paroxysmal AF patients. Furthermore, serum anti-M2 antibodies levels correlated with LAD and plasma NT-proBNP levels. The preprocedural level of anti-M2 was proved to be an independent predictor for the recurrence of AF at one-year follow-up after ablation [35].

Furthermore, it was demonstrated that the anti-M2 antibodies level was significantly increased in patients with lone paroxysmal AF when compared to healthy controls. It was also proved that, in the case of lone paroxysmal AF patients, serum anti-M2 antibodies level was a marker of left atrial fibrosis and thus an independent predictor of late AF recurrence [36]. 

Ma et al. compared long-standing persistent AF patients undergoing hybrid ablation and patients in sinus rhythm scheduled to undergo coronary artery bypass grafting (CABG) surgery. The study revealed that the first group had higher anti-M2 antibody levels, which were found to be associated with the severity of atrial fibrosis. In addition, serum anti-M2 antibody levels were proved to be positively correlated with TGF-β1 and connective tissue growth factor (CTGF) expression in the left atrial appendage [37].

Patients suffering from lone paroxysmal AF had a higher prevalence of increased levels of anti-M2 antibodies compared to patients with both AF and AH. However, there was no difference in the prevalence of increased levels of anti-M2 immunoglobulin M (IgM) between these groups [38].

Deng et al. used an animal model with two groups: (i) six rabbits immunized with M2 and treated with T4, and (ii) four rabbits treated only with T4. Increased inducibility of AF and sinus tachycardia was found in the first group and most likely it was gained through the disruption of electrophysiological properties mediated by anti-M2 antibodies and T4 collectively [39]. All studies discussed in this section have been summarized in Table 2.

### 2.3. Research Investigating Both Anti-β and Anti-M2 Antibodies

Stavrakis et al. studied Graves’ hyperthyroidism patients with AF or in sinus rhythm. It was shown that anti-β1 and anti-M2 antibodies were the strongest predictors of AF occurrence. Moreover, IgG electrophysiologic effects were analyzed using intracellular recordings from isolated canine pulmonary veins. It was shown that these antibodies induced hyperpolarization, decreased action potential duration, enhanced early afterdepolarization formation, and facilitated triggered firing in pulmonary veins by the local autonomic nerve stimulation [40]. Another research group evidenced that anti-β1 and anti-M2 antibody levels were determinants of AF occurrence in Graves’ disease patients [41]. Moreover, Yalcin et al. showed that anti-β1 and anti-M2 antibodies were independent biomarkers of lone paroxysmal AF presence in patients without concomitant cardiovascular diseases [42]. 

It was proved that preprocedural anti-β1 and anti-M2 antibody levels were independent predictors of late AF recurrence following cryoballoon-based pulmonary vein isolation in paroxysmal AF patients [43]. Hu et al. measured the serum levels of anti-β1 and anti-M2 antibodies, antinuclear antibodies, interleukin-6 (IL-6), and CRP in patients with non-valvular AF. It was demonstrated that left atrium diameter and the levels of anti-β1 antibodies, anti-M2 antibodies, and IL-6 were independent risk factors for non-valvular AF [44].

Li et al. immunized rabbits with peptides from the extracellular loops of both β1-adrenergic receptors and M2-muscarinic receptors to produce both types of antibodies. Antibodies expression triggered sustained sinus, junctional and atrial tachycardias, but not AF. Sustained AF was induced by addition of excessive T4. Interestingly, AF induction was blocked by the neutralization of these antibodies despite continued hyperthyroidism [45]. 

It was found that when considering patients with hypertrophic cardiomyopathy (HCM), the subgroup of patients with AF had remarkably higher concentrations of anti-β1 and anti-M2 antibodies than the subgroup without this disease. Atrial fibrillation in HCM patients led to atrial systolic dysfunction and lowered cardiac output in addition to ventricular diastolic dysfunction, leading to further deterioration of heart failure and elevation of anti-β1 and anti-M2 antibodies concentration [46]. All studies discussed in this section have been summarized in Table 3.

### 2.4. Research Investigating Anti-HSP Antibodies

The heat shock protein 65 (HSP65) is known to be expressed on the cell membrane only under stressful conditions. When HSP65 is attacked by circulating anti-HSP65 antibodies, it can cause a damaging autoimmune response mediated by complement-dependent cytotoxicity, leading to myocyte injury, electro-anatomical substrate formation, and in turn AF. Mandal et al. investigated the association between the preoperative levels of anti-HSP65 autoantibodies and the occurrence of postoperative AF in 329 patients undergoing elective primary CABG. It was confirmed that there was a positive association between anti-HSP65 antibody levels and postoperative AF occurrence independent of age, sex, or any other cardiovascular risk factors. Interestingly, preoperative CRP levels were similar in patients who had postoperative AF and in patients who did not develop this tachyarrhythmia [47].

Oc et al. researched the difference between preoperative and postoperative levels of anti-HSP60 antibodies in 20 patients undergoing elective primary CABG. The study revealed that increased preoperative levels of anti-HSP60 antibodies positively correlated with the development of postoperative AF [48]. 

The same research group in another prospective study proved that anti-HSP70 antibodies were a predictor of AF development in patients undergoing the CABG procedure. In addition, there was no significant difference in the serum CRP levels between the study and the control group [49].

Kornej et al. investigated changes in the levels of HSP70 and anti-HSP70 antibodies as well as rhythm outcomes in 67 patients with AF undergoing catheter ablation. Anti-HSP70 antibody levels were proved to be associated with AF type: patients with persistent AF had higher anti-HSP70 antibody titers than those with paroxysmal AF. Moreover, the level of HSP70 and anti-HSP70 antibodies increased after AF catheter ablation. Interestingly, the HSP70 level increase was associated with total ablation time and energy. Furthermore, in univariate analysis, the growth in the level of HSP70 and anti-HSP70 antibodies was associated with a higher AF recurrence rate during 6 months follow-up after the ablation procedure [50]. All studies discussed in this section have been summarized in Table 4.

### 2.5. Research Investigating Other Types of Autoantibodies

Patients with idiopathic paroxysmal AF had an increased prevalence of anti-cardiac myosin heavy chain (anti-cMHC) IgG as compared to healthy controls with sinus rhythm. Moreover, all AF patients with anti-cMHC IgG had also specific reactivity in their sera against both ventricular and atrial cardiac MHC isoforms [51].

Baba et al. compared the levels of anti-myosin, anti-β_1,_ and anti-Na^+^-K^+^-ATPase antibodies between patients suffering from heart failure with reduced ejection fraction and control patients with hypertension alone. They discovered that antibodies against Na^+^-K^+^-ATPase were an independent risk factor for the occurrence of paroxysmal AF in congestive heart failure patients [52]. Another research group undertook a study to find out whether anti-serum amyloid A (SAA) antibodies could be detected in a group of AF patients and whether these antibodies were associated with inflammatory states, using CRP as an inflammation marker. They discovered the association between elevated levels of anti-SAA antibodies and the occurrence of AF. Interestingly, there was no association between anti-SAA levels and CRP levels [53].

Furthermore, Xu et al. demonstrated that among patients with cardiovascular manifestations of Graves’ disease, the frequency of AF was significantly higher in patients who were anti-angiotensin II-1 receptor (anti-AT_1_) antibodies positive compared to those who were anti-AT_1_ antibodies negative [54].

Blagova et al. analyzed anti-heart antibodies in 34 coronavirus disease 2019 (COVID-19) pneumonia patients. Anti-smooth muscle antibodies (ASMA) levels correlated with the presence of AF at admission and the need for oxygen therapy [55].

Moreover, it was shown that in the group of patients with acute ischemic stroke, patients with positive initial antiphospholipid antibodies (lupus anticoagulant screening, anti-cardiolipin antibody or anti-β2glycoprotein I antibodies) had more AF episodes and higher levels of CRP than patients without these antibodies [56]. Importantly, Sveen et al. studied the incidence of AF and the levels of anti-apoB100 antibodies (meaningfully against peptides p45 and p210) in a large population-based cohort. It was shown that increased levels of IgM against apoB100 p210 were independently associated with a decreased risk of AF development but only in women. Moreover, the association between IgM and AF was significantly different between sexes even after adjusting risk factors and comorbidities [57]. All studies discussed in this section have been summarized in Table 5.

## 3. Conclusions

The significance of autoimmune processes and particular autoantibodies in the development of AF has been the field of extensive research. Serum concentrations of various autoantibodies have been proven to be significantly increased in patients with different types of AF when compared to healthy people. Most scientific attention has been applied to activating autoantibodies against receptors of the autonomous nervous system. Moreover, a growing number of studies has shown that other autoantibodies may affect pathogenetic pathways leading to AF (Figure 2). Importantly, there are many gaps in evidence concerning mainly the cause-and-effect relationships and, therefore, observed correlations do not indicate the unambiguous role of autoantibodies in AF development and they need to be further investigated.

Noteworthily, not only have concentrations of various antibodies and their correlation with specific types of AF been investigated, but also their predictive values, their correlations with other concomitant heart conditions, and possible pathophysiological mechanisms that determine their impact on atrial structural and electrophysiological changes. Nevertheless, the exact role of various autoantibodies in AF is still a matter of ongoing debate.

## 4. Future Perspectives

We propose that inflammatory cytokines and immune cells should be evaluated in future investigations concerning autoantibodies in AF. The exact characteristics of immune response could contribute to a better understanding of autoantibodies’ roles in the autoimmune process and therefore cause-and-effect relationships could be demonstrated. The identification of immunological autoantibody-mediated mechanisms would reveal new perspectives in the treatment and prevention of AF, including the use of immunosuppressive agents or neutralizing antibodies. Moreover, autoantibodies could potentially be used as a screening tool to determine the risk of developing AF in the future, especially in patients with underlying cardiovascular system diseases. 

Further investigations in larger prospective studies are needed to determine specific autoantibodies level profiles as a pre-ablation screening tool to spot patients likely to have arrhythmia recurrence, and thus to contribute to the improvement of the efficacy of this treatment method. It would lead to a more individualized therapy with an estimated better risk-benefit ratio.

Additionally, there remains a substantial need to establish cut-off values to use the discussed circulating antibodies as a screening tool for patients with different structural heart diseases to prevent AF evolution. 

Finally, as recent evidence suggests that neutralization of anti-β1 and anti-M2 antibodies blocks induction of AF, it is warranted to undertake studies, both in vivo and in vitro, investigating the use of decoy peptide therapy. Observations made in these studies could then be translated into clinical medicine in the form of new treatment regimens.

## Figures and Tables

**Figure 1 ijms-24-01852-f001:**
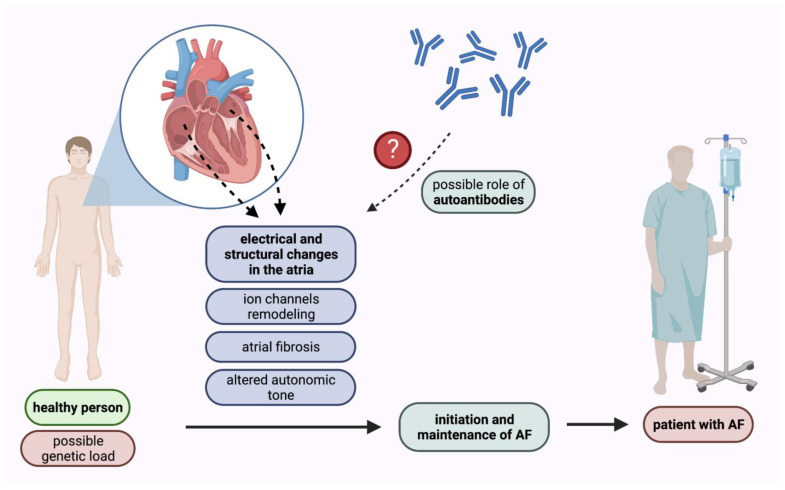
Pathomechanisms of AF initiation and maintenance with the possible role of autoantibodies. AF—atrial fibrillation.

**Figure 2 ijms-24-01852-f002:**
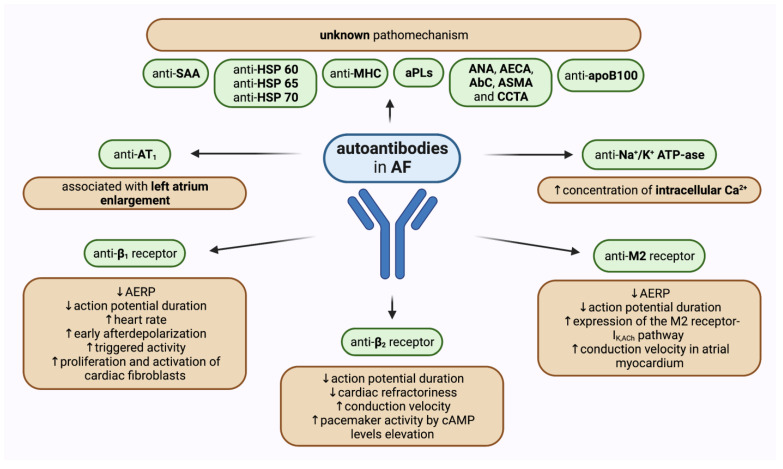
Graphical summary of the role of different autoantibodies in atrial fibrillation with a probable pathomechanism of action if known. ↑—increased, ↓—decreased, AbC—anti-cardiomyocyte antibodies, AECA—anti-endothelial cell antibodies, AERP—atrial effective refractory period AF—atrial fibrillation, ANA—anti-nuclear antibodies, aPLs—antiphospholipid antibodies, ASMA—anti-smooth muscle antibodies, AT_1_—angiotensin II receptor type 1, β_1_—β_1_-adrenergic receptors, β_2_—β_2_-adrenergic receptors, cAMP—cyclic adenosine monophosphate, CCTA—cardiac conducting tissue antibodies, HSP—heat shock protein, I_K,Ach_—muscarinic-gated potassium channel, M2—M2-muscarinic acetylcholine receptor, MHC— myosin heavy chain, SAA—serum amyloid A.

**Table 1 ijms-24-01852-t001:** Summary of recent research investigating only antibodies against β-adrenergic receptors.

Ref.	Year	Population	Comparison	Outcome	Methodology	Type of Study
[27]	2004	110 VT/VE/AF pts (59 primary arrhythmias, 33 CM and DCM, 18 IHD)	20 healthy controls	↑ anti-β_1_ Abs levels in VT/VE/AF pts (all groups excluding IHD) compared to controls and IHD pts	anti-β_1_ Abs in serum by direct immunoassay	clinical
[28]	2006	(i) 35 idiopathic arrhythmias pts (AF or flutter and AT) (ii) 15 SVA and DCM or CM pts (iii) 23 SVA and AH pts	20 healthy controls	↑ prevalence of anti-β_1_ Abs in arrhythmic pts compared to controls	anti-β_1_ Abs in serum by direct immunoassay	clinical
[29]	2006	42 arrhythmic pts: (i) 31 idiopathic (13 paroxysmal AF or flutter, 2 PAT and 16 PVT pts) (ii) 11 PVT and DCM or CM pts	20 healthy controls	↑ prevalence of anti-β_1_ Abs in arrhythmic pts compared to controls	anti-β_1_ Abs in serum by direct immunoassay	clinical
[30]	2013	5 rabbits immunized with β_2_ after immunization	the same 5 rabbits before immunization	↑ anti-β_2_ Abs levels lead to atrial arrhythmias (mainly SAT)	anti-β_2_ Abs in serum by ELISA	preclinical
[31]	2014	9 rabbits immunized with β_1_ (*n* = 5) or β_2_ (*n* = 4) and injected with T4 4 rabbits injected with T4 alone	the same 9 rabbits but before immunization	SST induction by T4 and anti-β_1_ Abs SAT induction by T4 and anti-β_2_ Abs SJT and SST by T4 alone	anti-β_1_ Abs and anti-β_2_ Abs in serum by ELISA	preclinical
[32]	2020	70 paroxysmal AF pts	correlation between anti-β_1_ Abs levels and other parameters	positive correlation of anti-β_1_ Abs levels with LAD and circulating fibrosis biomarker levels	anti-β_1_ Abs and circulating fibrosis biomarkers in serum by ELISA	clinical
[32]	2020	8 rabbits immunized with β_1_ ECL2 peptide	8 rabbits not immunized	↑ HR, AF inducibility; LAD, systolic dysfunction, and fibrosis of LA in immunized rabbits	anti-β_1_ Abs in serum by ELISA	preclinical

↑—increased, Abs—antibodies, AF—atrial fibrillation, AH—arterial hypertension, AT—atrial tachycardia, β_1_—β_1_-adrenergic receptors, β_2_—β_2_-adrenergic receptors, CM—chronic myocarditis, DCM—dilated cardiomyopathy, ECL2—second extracellular loop, ELISA—enzyme-linked immunosorbent assay, HR—heart rate, IHD—ischemic heart disease, LA—left atrium, LAD—left atrial diameter, PAT—paroxysmal atrial tachycardia, pts—patients, PVT—paroxysmal ventricular tachycardia, ref.—reference, SAT—sustained atrial tachycardia, SJT—sustained junctional tachycardia, SST—sustained sinus tachycardia, SVA—supraventricular arrhythmia, T4—thyroxine, VE—ventricular extrasystole, VT—ventricular tachycardia.

**Table 2 ijms-24-01852-t002:** Summary of recent research investigating only antibodies against M2-muscarinic acetylcholine receptor.

Ref.	Year	Population	Comparison	Outcome	Methodology	Type of Study
[33]	2004	104 DCM pts and 104 idiopathic AF pts	104 healthy controls	anti-M2 Abs as an independent predictor of the presence of AF in DCM pts	anti-M2 Abs in serum by ELISA	clinical
[33]	2004	chick embryos injected with IgG from: (i) DCM pts with anti-M2 Abs, and (ii) idiopathic AF pts with anti-M2 Abs	chick embryos injected with IgG from: (i) healthy controls, (ii) DCM pts without anti-M2 Abs, and (iii) idiopathic AF pts without anti-M2 Abs	anti-M2 Abs from AF and DCM pts exhibited negative chronotropic effects and induced SVAs	IgG purified by Protein A column	preclinical
[34]	2009	16 rabbits immunized with M2	16 rabbits injected with saline	↑ repetitive atrial responses, expression of the M2 receptor-IK,ACh pathway and atrial fibrosis in M2 immunized rabbits compared to rabbits injected with saline	anti-M2 Abs in serum by ELISA	preclinical
[35]	2013	76 AF pts (43 paroxysmal and 33 persistent) subjected to CAPV	77 NSR pts (AH or CAD or medical checkup)	↑ prevalence and levels of anti-M2 Abs in AF group compared to controls ↑ prevalence and levels of anti-M2 Abs in persistent AF pts compared to paroxysmal AF pts pre-procedural level of anti-M2 Abs as an independent predictor for the recurrence of AF one year after CAPV	anti-M2 Abs in serum by ELISA	clinical
[36]	2015	31 paroxysmal lone AF pts after cryoballoon ablation	31 healthy controls	↑ anti-M2 Abs in paroxysmal lone AF pts compared to healthy controls anti-M2 Abs as a marker of LA fibrosis LA fibrosis as an independent predictor of late AF recurrence	anti-M2 Abs in serum by ELISA	clinical
[37]	2019	24 persistent AF pts undergoing HA	26 NSR pts, undergoing CABG	↑ anti-M2 Abs levels in AF pts associated with the severity of atrial fibrosis	anti-M2 Abs in serum by ELISA	clinical
[38]	2019	100 lone AF pts	84 AF AH pts	↑ prevalence of anti-M2 Abs in paroxysmal AF pts compared to AF AH pts ↑ levels of anti-M2 IgG but not IgM in lone AF pts	anti-M2 IgG and IgM in serum by indirect immunoenzyme assay	clinical
[39]	2021	Group A: 6 rabbits immunized with M2 and treated with T4 Group B: 4 rabbits treated only with T4	rabbits before immunization and before treatment	↑ prevalence of sustained ST and AF by T4 and immunization in comparison to the preimmunization state	anti-M2 Abs in serum by ELISA	preclinical

↑—increased, Abs—antibodies, AF—atrial fibrillation, AH—arterial hypertension, CABG—coronary artery bypass grafting, CAD—coronary artery disease, CAPV—circumferential ablation of pulmonary vein, DCM—dilated cardiomyopathy, ELISA—enzyme-linked immunosorbent assay, HA—hybrid ablation, I_K,Ach_—muscarinic-gated potassium channel, Ig—immunoglobulin, LA—left atrium, M2—M2-muscarinic acetylcholine receptor, NSR—normal sinus rhythm, pts—patients, ref.—reference, ST—sinus tachycardia, SVA—supraventricular arrhythmia, T4—thyroxine.

**Table 3 ijms-24-01852-t003:** Summary of recent research investigating both antibodies against β-adrenergic receptors and antibodies against M2-muscarinic acetylcholine receptor.

Ref.	Year	Population	Comparison	Outcome	Methodology	Type of Study
[40]	2009	17 GD pts with AF	21 GD pts with NSR	anti-β_1_ Abs and anti-M2 Abs as strong predictors of AF no difference between sexes	anti-β_1_ Abs and anti-M2 Abs in serum by ELISA	clinical
[41]	2015	31 GD AF pts 36 GD NSR pts 9 TMG pts 5 SAT pts	10 healthy controls	anti-β_1_ Abs and anti-M2 Abs as risk factors for AF in GD pts no difference between sexes ↓ prevalence of Abs in TMG and SAT pts	anti-β_1_ Abs, anti-β_2_ Abs, and anti-M2 Abs in serum by ELISA	clinical
[43]	2015	80 paroxysmal AF pts with preserved LVEF subjected to cryoballoon PVI	the same pts at 1-year follow-up after ablation	preprocedural anti-β_1_ Abs and anti-M2 Abs levels as independent predictors of late AF recurrence following cryoballoon PVI	anti-β_1_ Abs and anti-M2 Abs in serum by ELISA	clinical
[42]	2015	75 lone paroxysmal AF pts	75 healthy controls	anti-β_1_ Abs and anti-M2 Abs as independent prognostic factors of lone paroxysmal AF	anti-β_1_ Abs and anti-M2 Abs in serum by ELISA	clinical
[44]	2016	71 non-valvular AF pts	75 healthy controls	anti-β_1_ Abs and anti-M2 Abs levels as independent prognostic factors of non-valvular AF	anti-β_1_ Abs and anti-M2 Abs in serum by ELISA	clinical
[45]	2016	5 rabbits immunized with β_1_ and M2 + addition of excessive T_4_	the same 5 rabbits before immunization	sustained AF induction after T_4_ injection to immunized rabbits AF induction blocked by neutralization of Abs	anti-β_1_ Abs and anti-M2 Abs in serum by ELISA	preclinical
[46]	2020	134 HCM pts	40 healthy subjects	↑ levels of anti-β_1_ Abs and anti-M2 Abs in AF pts compared to pts without AF ↑ levels of anti-M2 Abs in women compared to men	anti-β_1_ Abs and anti-M2 Abs in serum by ELISA	clinical

↑—increased, ↓—decreased, Abs—antibodies, AF—atrial fibrillation, β_1_—β_1_-adrenergic receptors, ELISA—enzyme-linked immunosorbent assay, GD—Graves’ disease, HCM—hypertrophic cardiomyopathy, LVEF—left ventricular ejection fraction, M2—M2-muscarinic acetylcholine receptor, NSR—normal sinus rhythm, pts—patients, PVI—pulmonary vein isolation, ref.—reference, SAT—subacute thyroiditis, T4—thyroxine, TMG—toxic multinodular goiter.

**Table 4 ijms-24-01852-t004:** Summary of recent research investigating antibodies against heat shock proteins.

Ref.	Year	Population	Comparison	Outcome	Methodology	Type of Study
[47]	2004	62 CABG undergoing pts with postoperative AF	267 CABG undergoing pts without postoperative AF	↑ prevalence of anti-HSP65 Abs in pts with postoperative AF no difference between sexes	anti-HSP65 Abs in serum by ELISA	clinical prospective
[48]	2007	10 CABG undergoing pts with postoperative AF	10 CABG undergoing pts without postoperative AF	↑ anti-HSP60 Abs levels in pts with postoperative AF	anti-HSP60 IgG in serum by ELISA	clinical prospective
[49]	2008	10 CABG undergoing pts with postoperative AF	10 CABG undergoing pts without postoperative AF	↑ anti-HSP70 Abs levels in pts with postoperative AF	anti-HSP70 Abs in serum by ELISA	clinical prospective
[50]	2013	67 AF (paroxysmal or persistent) pts undergoing ablation	34 healthy controls	↑ anti-HSP70 Abs levels in persistent AF pts compared to paroxysmal AF pts HSP70 levels associated with total ablation time and energy ↑ AF recurrence rate in pts with ↑ anti-HSP70 Abs levels	anti-HSP70 Abs and HSP70 in plasma by ELISA	clinical prospective

↑—increased, Abs—antibodies, AF—atrial fibrillation, CABG—coronary artery bypass grafting, HSP—heat shock protein, Ig—immunoglobulin, pts—patients, ref.—reference.

**Table 5 ijms-24-01852-t005:** Summary of recent research investigating other types of autoantibodies.

Ref.	Year	Population	Comparison	Antibody	Outcome	Methodology	Type of Study
[51]	1998	10 idiopathic paroxysmal AF pts	10 healthy controls	anti-cMHC IgG	↑ prevalence of anti-MHC IgG in AF pts compared to controls	Abs in serum by SDS-PAGE and Western blotting	clinical
[52]	2002	95 HFrEF pts (48 CAD, 47 DCM)	48 control hypertensive patients	anti-M Abs anti-β_1_ Abs anti-NKA Abs	anti-NKA Abs as an independent risk factor for the occurrence of paroxysmal AF no difference between sexes	anti-M Abs by immunofluorescence anti-β_1_ Abs and anti-NKA Abs in serum by ELISA	clinical
[53]	2004	13 AF pts	62 healthy controls	anti-SAA Abs	↑ anti-SAA Abs levels in pts with AF compared to healthy controls	Abs in serum by ELISA	clinical
[54]	2014	118 GD pts divided into groups with or without cardiovascular manifestations	40 healthy controls	anti-AT_1_ Abs	↑ AF frequency in the anti-AT_1_ Abs positive group compared to the anti-AT_1_ Abs negative group (within pts with cardiovascular manifestations of GD)	Abs in serum by ELISA	clinical
[55]	2021	34 COVID-19 pneumonia pts	analysis within the studied population: AHA levels comparison and their correlation with clinical outcomes	ANA, AECA, AbC, ASMA, and CCTA	correlation between ASMA levels and occurrence of AF	Abs in serum by enzyme immunoassay	clinical
[56]	2021	739 patients with acute ischemic stroke	analysis within the studied population: comparison of Abs levels between pts	aPLs	↑ levels of CRP and AF occurrence in pts with positive initial aPLs	lupus anticoagulant by dRVVT anti-cardiolipin and anti-β_2_-glycoprotein Abs in serum by ELISA	clinical
[57]	2022	5169 individuals from the Malmö Diet and Cancer cohort	analysis within the studied population: correlation between anti-apoB100 Abs levels and AF incidence	IgM and IgG against apoB100 peptides (p45 and p210)	↓ risk of AF in females with ↑ levels of IgM against p210	Abs in serum by ELISA	clinical

↑—increased, ↓—decreased, AbC—anti-cardiomyocyte antibodies, Abs—antibodies, AECA—anti-endothelial cell antibodies, AF—atrial fibrillation, AHA—anti-heart antibodies, ANA—anti-nuclear antibodies, aPLs—antiphospholipid antibodies, ASMA—anti-smooth muscle antibodies, AT_1_—angiotensin II receptor type 1, β_1_—β_1_-adrenergic receptors, CAD—coronary artery disease, CCTA—cardiac conducting tissue antibodies, cMHC—cardiac myosin heavy chain, COVID-19—coronavirus disease 2019, CRP—C-reactive protein, DCM—dilated cardiomyopathy, dRVVT—diluted Russel’s viper venom time, ELISA—enzyme-linked immunosorbent assay, GD—Graves’ disease, HFrEF—heart failure with reduced ejection fraction, Ig—immunoglobulin, M—myosin, NKA—Na^+^-K^+^-ATPase, pts—patients, ref.—reference, SAA—serum amyloid A, SDS-PAGE—sodium dodecyl sulfate polyacrylamide gel electrophoresis.

## Data Availability

Not applicable.

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
