# Peer review of "Autoantibodies in Atrial Fibrillation—State of the Art"

_ijms, 2023, doi:10.3390/ijms24031852_

Round 1

Reviewer 1 Report

In this work, Zygadlo et al aim to summarize the current knowledge regarding the involvement of autoantibodies in atrial fibrillation (AF) development and their prognostic value in the disease. AF is a multifactorial disorder and participation of the autoimmune disease in AF progression is a relatively new field. The topic of research is timely and important. Although there are few recent revisions of the participation of autoantibodies in cardiac arrhythmias, the authors provided a structured analysis of the current knowledge of the field. Overall, Zygadlo et al provided a good amount of information to overview the role of several different autoantibodies in AF. I have few concerns about the data presentation.

Specific concerns:

1.   In the introduction, the authors described the autoimmunity and the general mechanisms of AF. Authors are encouraged to provide the more specific introduction to the general action of the autoantibodies. How can they affect the target protein/receptor function?

2.   There were 31 original research works included in the manuscript. Please provide information if there were any work related to the topic excluded. What were the criteria for the exclusion?

3.    Authors summarize the findings in the tables to make the information clear. However, they highlight the sex-related difference of antibodies only for one research work. Please provide information on sex-related differences for the rest of the works where it was shown.

4.   In a few preclinical works described in the manuscript, the antibodies were introduced along with T4. Please briefly describe the possible role of T4 in autoimmune response and its relation with the autoantibodies.

Minor concerns.

The manuscript requires additional proofreading. There are some abbreviations introduced several times or described after the abbreviation place. For example, HSP65 (line207) and heat shock protein (line 211), “adrenoreceptor” and “AR” are exchanged multiple times within the text.

Table 5, papers 55-57: there is no control patient information in the Comparison column. 

Reviewer 2 Report

The manuscript of Zygadlo et al presents substantial information regarding the presence of autoantibodies against beta1 and beta2 adrenoceptors as well as against M2 cholinergic receptors or heat shock proteins in patients with different forms of atrial fibrillation. They nicely discuss the link between these autoantibodies and the AF initiation and perpetuation from the point of view of experimental animals subjected to immunization with one or more peptides against adrenoceptors or muscarinic receptors. They carefully compare these findings with the findings in human patients. The article is in general well written, however, there are minor but important aspects that should be corrected:

-in general, for the cited studies where the groups of patients are very small, like in refs 28, 29 or 48, 49, published between 2006 to 2008, were those findings confirmed in later reports? 

- it has never been explained that the term  anti-β antibodies refers to antibodies against beta adrenergic receptors . This is mostly happening for β1 adrenergic receptors, while for β2-AR the explanation exists both in the text and in the legends of the tables. Please correct.

- the abbreviation AR for acetylcholine receptor is confusing, since the same abbreviation is used for adrenergic receptor in β2AR. Please use another abbreviation. The most scientific reports use AR for adrenergic receptors. The cholinergic muscarinic receptor type 2 is simply M2 or M2R (M2 receptor).

- It is not correct to write "anti-β1 and anti-M2-AR", since in this situation, AR refers to different types of receptors. This formulation exists in the whole manuscript.

-L 158: please decide between made and used

Reviewer 3 Report

Interesting comprehensive review of a novel idea of Ab related mechanism in the pathogenesis of AF. 
